# EDeR: Towards Understanding Dependency Relations Between Events

**Ruiqi Li♠, Patrik Haslum♠, Leyang Cui♡†**

♠ Australian National University
♡ Tencent AI lab
*{ruiqi.li, patrik.haslum}@anu.edu.au    leyangcui@tencent.com*

## Abstract

Relation extraction is a crucial task in natural language processing (NLP) and information retrieval (IR). Previous work on event relation extraction mainly focuses on hierarchical, temporal and causal relations. Such relationships consider two events to be independent in terms of syntax and semantics, but they fail to recognize the interdependence between events. To bridge this gap, we introduce a human-annotated **E**vent **De**pendency **R**elation dataset (EDeR). The annotation is done on a sample of documents from the OntoNotes dataset, which has the additional benefit that it integrates with existing, orthogonal, annotations of this dataset. We investigate baseline approaches for EDeR's event dependency relation prediction. We show that recognizing such event dependency relations can further benefit critical NLP tasks, including semantic role labelling and co-reference resolution.

## 1 Introduction

Events play a critical role in enabling AI agents to understand and perceive the world, as they provide information on what happened and the entities involved. An event is composed of a predicate (i.e., verb) and arguments, where the predicate indicates the event's action and the arguments represent the subject, object and so on of the predicate (Levin et al., 1999; Rappaport Hovav et al., 2010).

Previous work on relation extraction mainly focuses on investigating the relationships between entities (Miwa and Sasaki, 2014; Huguet Cabot and Navigli, 2021; Chen et al., 2020; Ma et al., 2022) rather than events. In the relatively limited research work that studies the relations between events, the types of relations considered are causal (Mariko et al., 2020, 2022; Tan et al., 2022), temporal (Bethard, 2013; Laokulrat et al., 2013), and hierarchical (Hovy et al., 2013; Glavaš and Šnajder, 2014). Such relationships consider two events

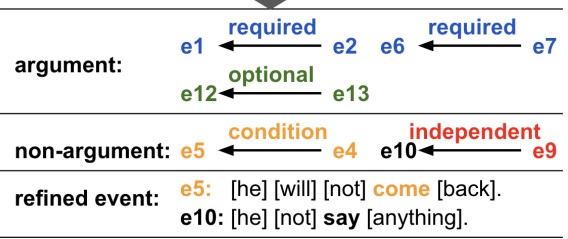

Figure 1: Examples of the event dependency relations. Above: Source text. (Event predicates are marked in boldface and argument spans are marked with brackets.) Below: Event dependency relations and refined events.

to be independent in terms of syntax and semantics but overlook the inter-dependence between them. This disregard for interdependence can potentially yield incomplete or erroneous narrative interpretations. Take the sentence "He tried to forgive his father one time." from Figure 1, $e2$ with the predicate "forgive" is the action that he "tried" ($e1$) to take. However, there is no temporal or causal or hierarchical relation between $e2$ and $e1$; rather, $e2$ is the object (syntactically depending) and patient (semantically depending) of $e1$. In fact, $e2$ didn't actually happen. Recognizing such relations, that reveal both semantic and syntactic dependencies, is an essential, yet challenging task for AI researchers. Correctly identifying these dependency relations can enhance language understanding and provide valuable information for various related NLP tasks.

Motivated by these, we investigate how an event may be an **argument** of another as representing the event dependency relations, rather than being regarded as **independent**. Additionally, we distinguish argument into **required argument** and

---

† Corresponding author.

**optional argument** events. Required arguments, like $e2$ ("forgive") in $e1$ ("tried") from Figure 1, are present for an event to be complete and meaningful (to "try to", one has to try to do something). Optional arguments, such as $e13$ ("see") in $e12$ ("carried"), enhance or clarify the event that it is an argument of (for example indicating the purpose), and their occurrence isn't explicitly stated. Non-argument events, which are conjunctive or conditional, are often mislabelled as arguments in the event extraction task, like $e9$ ("arrived"), a temporal modifier for $e10$ ("did not [say] anything"), but is independent.

Argument-event dependency relations are frequent in natural language texts, especially in narrative texts such as news articles, conversations, and similar. Hence, we present a human-annotated **E**vent **De**pendency **R**elation dataset (EDeR) that (1) extracts event dependency information based on a sample of 275 documents spanning seven genres from OntoNotes (Pradhan et al., 2013) and integrates with orthogonal annotations of this dataset, and (2) provides refined semantic role-labelled event representations based on this information. We build the EDeR dataset with 11,852 high-quality annotations, according to the proposed event relation taxonomy. It can serve as the basis for an effective automatic classification process.

We apply both EDeR human-annotated and model-predicted event dependency relations to enhance a state-of-the-art semantic role labeling (SRL) model (Zhang et al., 2022)'s event representation generation. Experimental results prove these relations improve event extraction, yielding updated representations from EDeR. These refined event representations were further applied to the co-reference resolution (CR) task. A performance comparison taking the original and our updated event representations as inputs on a CR model (Jiang and Cohn, 2021) affirms the effectiveness and validity of EDeR's refined representations. We release EDeR (e.g., dataset, baseline code, and model) at `https://github.com/RichieLee93/EDeR`.

## 2 Background and Related Work

### 2.1 Event and Event Representation

Event mentions involve a predicate (usually a verb or phrasal verb) and a set of labelled arguments (Levin et al., 1999; Rappaport Hovav et al., 2010). Events can be identified from texts using

| datasets | reps. | | | relations |
|---|---|---|---|---|
| | verb | arg. | span | |
| ECB (Bejan and Harabagiu, 2008) | ✓ | ✗ | ✗ | H |
| Hieve (Glavaš et al., 2014) | ✓ | ✗ | ✗ | H |
| IC (Araki et al., 2014) | ✓ | ✗ | ✗ | H |
| TimeBank (Pustejovsky et al., 2003) | ✓ | ✗ | ✗ | T |
| TB-Dense (Cassidy et al., 2014) | ✓ | ✗ | ✗ | T |
| MATRES (Ning et al., 2018b) | ✓ | ✗ | ✗ | T |
| ECD (Do et al., 2011) | ✓ | ✗ | ✗ | CA |
| CiRA (Fischbach et al., 2021) | ✗ | ✗ | ✓ | CA |
| CNC (Tan et al., 2022) | ✗ | ✗ | ✓ | CA |
| FCR (Yang et al., 2022) | ✗ | ✗ | ✓ | CA |
| CATENA (Mirza and Tonelli, 2016) | ✓ | ✗ | ✗ | CA+T |
| ESC (Caselli and Vossen, 2017) | ✓ | ✗ | ✗ | CA+T |
| TCR (Ning et al., 2018a) | ✓ | ✗ | ✗ | CA+T |
| EDeR (Ours) | ✓ | ✓ | ✓ | CO+D |

Table 1: Comparisons of event representations (reps.) and relations between our EDeR dataset and other public event relation datasets. "arg." is the abbreviation of argument labels. In the column "relations", "H", "T", "CA", "CO" and "D" represent hierarchical, temporal, causal, conditional and dependency, respectively.

semantic role labelling (SRL) which marks roles (i.e., predicates and arguments) of words or word spans in sentences. The OntoNotes dataset uses the PropBank annotation schema (Bonial et al., 2012) for SRL, categorizing arguments as text spans, assigning numbered arguments (ARG0-ARG5), and labelling verb modifiers like purpose (PRP) and location (LOC). Examples are shown in Figure 2. However, as Table 1 shows, most of the public event relation datasets, unlike EDeR, solely represent events as text verbs or spans without identifying arguments and argument labels. This could result in considerable information loss and increased ambiguity in the process of understanding an event.

We find that events can be arguments of other events, though not all events within another's span are arguments. This discrepancy can lead to shared subjects or objects without signifying an argument relationship. For instance, in Figure 2, the event "The man [who] works here" is contained in the event "The man who works here tells me to get out" but is not an argument of "tells". These annotations in the SRL datasets such as OntoNotes overlook the dependency relationship between events, but clarifying this relationship allows for a more focused argument span selection, such as "The man" for ARG0 (agent) of the event with the predicate "tells" in the example above.

### 2.2 Relations Between Events

Several event-event relations have been proposed in recent. The summary of varied relations of the

existing datasets is shown in Table 1's last column. The TimeBank (Pustejovsky et al., 2003), TB-Dense (Cassidy et al., 2014) and MATRES (Ning et al., 2018b) datasets focus on the temporal relationship which reveals if one event happens BEFORE, AFTER, etc. another. Logical relationships like causality are also explored in datasets like ECD (Do et al., 2011), ESC (Caselli and Vossen, 2017), CiRA (Fischbach et al., 2021) and CNC (Tan et al., 2022). Datasets like CATENA (Mirza and Tonelli, 2016), ESC (Caselli and Vossen, 2017), and TCR (Ning et al., 2018a) individually detect temporal and causal relations based on identical events. Such relationships regard two events as syntactically and semantically independent - the occurrence and actuality of each event are isolated and not affected by others. Besides, several datasets propose hierarchical relationships with sub-events satisfying temporal and spatial conditions within a super-event (Hovy et al., 2013; Glavaš et al., 2014; Glavaš and Šnajder, 2014). Bejan and Harabagiu (2008) and Araki et al. (2014) explore event coreference relations - whether two event mentions refer to the same event. Such hierarchical relations cannot deal with cases when an event (especially the verb of the event) requires a clausal complement, i.e., an argument of the event verb is itself an event.

# 3 Dataset

We utilize a subset of OntoNotes documents with its human-annotated and predicate-argument formatted events to extract candidate event pairs, where one event's predicate is contained in the span of another. These pairs, denoted as $Event1$ and $Event2$, are labelled by human annotators to indicate whether $Event2$ is a required argument of, an optional argument of, a condition of, or independent from $Event1$. We next detail the way we collect and pre-process the candidate event pairs and the human annotation process and construction of the dataset.

## 3.1 Data Collection

OntoNotes contains semantic role-formatted event representations, as the OntoNotes example in Figure 2 shows. We randomly sampled 275 documents from seven genres: broadcast news (bn), magazine (mz), newswire (nw), pivot corpus (pt), telephone conversation (tc), broadcast conversation (bc), and web data (wb). Detailed data statistics are shown

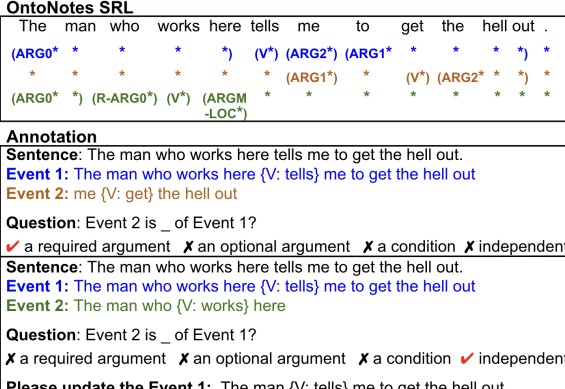

Figure 2: Above: A sample sentence with semantic role labels from the OntoNotes dataset. Below: Corresponding event pairs presented for human annotation with event dependency relations.

in A.1 Table 6. The number of sampled documents and the separation of them into training, development and test sets under each genre follows their initial distributions in the OntoNotes dataset.

### 3.1.1 Candidate Event Pair Extraction

Because arguments are spans of text, part or all of an extracted event may lie within the argument of another event. If the verb (i.e., predicate) of $e_j$ is within an argument of $e_i$, we say $e_j$ is *contained* in $e_i$. This can be nested. Contained events are candidates for being argument events, but are not necessarily so. For example, in the top part of Figure 2, the event in green (**e3** = {ARG0: The man } {R-ARG0: who } {V: works } {ARGM-LOC: here }) is contained in the span of the event in blue (**e1** = {ARG0: The man who works here } {V: tells } {ARG2: me } {ARG1: to get the hell out }), but **e3** is not an argument of **e1**; however, the event in orange (**e2** = {ARG1: me } {V: get } {ARG2: the hell out }) is an argument of **e1**.

We select event pairs $(e_i, e_j)$ where $e_j$'s verb is contained within the span of an argument of $e_i$ as candidate event pairs for annotation.

### 3.1.2 Preprocessing

We apply three filters to the selected candidate event pairs. First, we filter out candidate event pairs $(e_i, e_j)$ in which $e_j$ does not have any arguments. This typically occurs when $e_j$ is a modal verb indicating the tense of the verb in $e_i$. Second, events in the OntoNotes dataset sometimes mistake adjectives for verbs (e.g., "given" in "at any given time"), so we use the words' POS tags to filter these out. The POS tags of the event predicates are

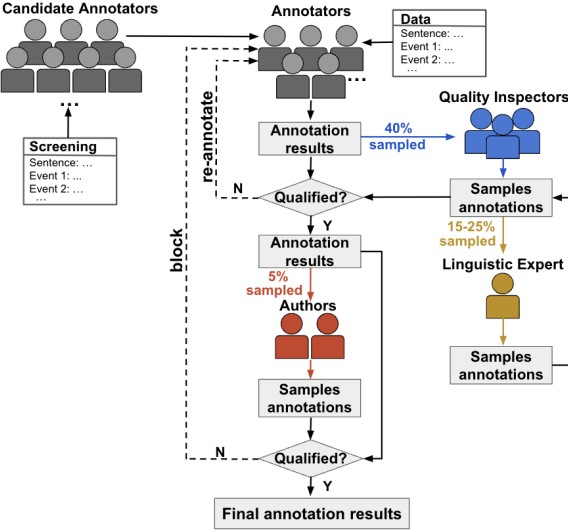

Figure 3: Illustration of our multi-level qualification-based annotation procedure.

obtained using the Stanford CoreNLP toolkit (Manning et al., 2014). Third, we remove transitively contained event pairs, i.e., pairs $(e_i, e_j)$ such that there exists another event $e_k$ such that the verb of $e_k$ is contained in $e_i$ and the verb of $e_j$ is contained in $e_k$. In such cases, $(e_i, e_k)$ and $(e_k, e_j)$ may be candidate pairs, but $(e_i, e_j)$ is not. For example, in "[She] {V: try } [to {V: stop } [[them] from {V: ruining } [...]]]"), "try" and "stop", and "stop" and "ruin" are two candidate pairs, but "try" and "ruin" are not.

## 3.2 Annotation

### 3.2.1 Annotation Instruction

We present each candidate pair with the whole span of both events and highlight the predicate of each event with "{V: }", as shown in the lower part of Figure 2. Annotators can choose one of four options to answer the relation between the two events. For the annotation task, we separate the non-argument case into two: *condition* and *independent* ("neither an argument nor a condition"). This made the definitions of the labels easier to understand, since a condition is also, like argument events, a hypothetical, rather than actual, event in the text. Identifying conditional statements in texts is itself an interesting and active topic of research (Fischbach et al., 2021; Tan et al., 2022), motivated in particular by their indication of causality. The full annotation instructions can be found in A.4.

### 3.2.2 Annotation Procedure

We adopt a multi-level qualification-based annotation procedure in which annotators' work is sampled and inspected/corrected in several stages, and feedback from the inspections is passed back to the annotators. A schematic of the procedure is shown in Figure 3. Compared with the commonly-used crowd-sourcing and voting strategy, this procedure makes annotators learn to improve throughout the task. Although in the end not all event pairs have been annotated by multiple participants, the strict qualification tests used along the way ensure annotators give their best answers.

Over 80 *candidate annotators* were recruited from colleges in China through a data annotation agency. They are English linguistic-major graduate students and fluent in English (passed the TEM-8 test[1]). After reading the instructions, candidates took a screening test, requiring them to annotate 50 examples, also annotated by authors, and cross-checked with the *linguistic expert*. The linguistic expert who has rich NLP-related annotation experience was hired via the same data annotation agency. 16 candidates who answered at least 85% of these questions correctly in the test were selected as *annotators*. Among the 16 annotators, the three with the highest accuracy were selected as *quality inspectors* (QIs). After further training, in which the linguistic expert explained the instructions and their mis-annotated cases from the screening test, the annotators started the annotation.

The event pairs for annotation were evenly split into 3 subsets and sequentially released to the annotators in 3 stages, each one week apart. In each stage, 40% of the annotators' submitted cases were randomly chosen and annotated by QIs. The expert reviewed 15-25% of QIs' inspected cases, provided feedback on errors, and offered explanations to prevent repetition in subsequent stages. This expert supervision ensures QIs' quality and prevents unreliable labels that could mislead annotators. After the linguistic expert's review of the QIs' annotations, if the agreement between an annotator's answers and the corresponding QI's is lower than 95%, the annotator must re-annotate inconsistencies until this agreement ratio is met.

The authors also monitor the quality of the annotation. In each stage, we sampled and labelled

---

[1]Test for English Majors-band 8, the highest level test in China for English major students that measures the overall English proficiency.

5% of the once-only annotated results from each annotator. If the accuracy achieved by an annotator is below 85%, the work of that annotator was discarded, and the annotator was dismissed from subsequent stages. Only one annotator failed this test and was removed after the first stage. Not counting the author-inspected cases, 4,771 (40.2% of all) cases received at least four times annotation.

### 3.2.3 Event Representation Refinement

As described in the second annotation example in Figure 2, one event (predicate) can be contained in another event but actually not an argument of it. For the event pairs that annotators label as independent, we further ask them to refine containing event ("Event 1") by removing the span of the contained event ("Event 2") from it. However, the contained event often shares the subject or object with the containing event, so annotators will keep any shared parts in Event 1 and remove the only remaining Event 2 spans. This manual event representation refinement proceeds simultaneously with the annotation of the event dependency relation and is inspected by the QIs and the linguistic expert as well.

Events tagged as conditions typically don't share subjects or objects with their enclosing events. We found automatic revision of the enclosing event feasible, hence didn't request annotators for it. If the conditional event $e_j$ and its containing event $e_i$ appear in the sentence as one of the subsequences (1) $e_i$ $s$ $e_j$ or (2) $s$ $e_j$ $e_i$ or (3) $e_j$, $e_i$, where $s$ is one of the signal words/phrases "if", "whenever", "as long as", "on [the] condition that", "unless", or "provided that", then we first remove all words in $e_i$ that are within the span of $e_j$, and, second, remove any signal word/phrase that is antecedent and adjacent to $e_j$. In the few cases when no signal word or phrase was detected in the second step, we manually checked and revised the containing event.

### 3.3 Analysis

Table 2 shows the distribution of event pairs classified under each label in the annotated dataset, as well as the number of event pairs in the training, development and test document subsets.

It is not surprising that the distribution is biased, with a majority (just over 75%) of event pairs labelled as "argument". The predicate-within-containing-event-span relation between events in the pairs we selected for annotation is a necessary condition for them to be dependent. Of the 2,901

pairs labelled as non-argument (condition or independent), 2,490 distinct events' representations are refined by the annotators and by our automated method.

| | argument | | non-argument | | overall | |
|---|---|---|---|---|---|---|
| | required | optional | cond. | indep. | | |
| **train** | 4,096 | 2,837 | 335 | 1,861 | 9129 | (77%) |
| **dev** | 635 | 421 | 41 | 355 | 1,452 | (12.3%) |
| **test** | 594 | 368 | 70 | 239 | 1271 | (10.7%) |
| **overall** | 5,325 (44.9%) | 3,626 (30.6%) | 446 (3.8%) | 2,455 (20.7%) | 11,852 | |

Table 2: Distribution of event pairs in the annotated EDeR dataset across labels and across the training, development and test subsets.

For human-annotated datasets, there is always a trade-off between the number of instances being annotated and the quality of annotations (Kryscinski et al., 2019; Cui et al., 2020). The size of our dataset is constrained by the annotation method. But it is comparable with or larger than many other human-annotated event relations reasoning datasets, e.g., Glavaš et al. (2014); Cassidy et al. (2014); Mirza and Tonelli (2016); Ning et al. (2018b); Tan et al. (2022). We conducted various kinds of inspections to ensure the quality of EDeR. The statistics are as follows: (1) Among the 4,771 samples (40.2% of the total 11,852 cases) from EDeR that have received the three Quality Inspectors' annotations, only 31 (0.65%) of them received three different labels from the three QIs; (2) we randomly sampled 417 cases that received at least four annotations; the accuracy of the annotated labels is 90.65% (378/417); (3) a 5.2% sample of the overall annotations was double-checked by the authors, with larger than 88% accuracy. These results show that the multi-level iterative annotation procedure produces high-quality annotation results.

## 4 Experiment: Event Dependency Relation Prediction

Given two events $e_i$ and $e_j$ from a sentence $X$, the basic task is to predict whether $e_j$ is an argument of $e_i$ or not. In this section, we evaluate the performance of several baseline methods on this task. The best achieves an accuracy of just over 82%. In Section 5, we show that even this binary classification of events into dependent and independent is sufficient to improve performance in two further NLP tasks: event representation extraction and co-reference resolution.

The task can be refined into a 3-way classification task, by distinguishing required and optional arguments, and into a 4-way classification task by distinguishing also the two types of non-argument events: conditions and independent events. The 3- and 4-way classification tasks are significantly harder. A summary of results is presented in Section 6, and full details in Appendix A.2.

## 4.1 Baseline Models

We evaluate several pre-trained language models, and a rule-based heuristic method.

The language models include discriminative models BERT (Devlin et al., 2019), Distil-BERT (Sanh et al., 2019), XLNet (Yang et al., 2019), RoBERTa (Zhuang et al., 2021); as well as two auto-regressive generative models GPT-2 (Radford et al., 2019) and ChatGPT[2] (i.e., GPT-3.5-turbo). Specifically, we feed ChatGPT four examples from the annotation instruction, representing each of the four relations used in the annotation as the prompt, along with the ground truth dependency labels, employing a "few-shot learning" approach. The implementation settings can be found in A.3.

We also design an unsupervised heuristic rules-based method method. The rules are: If any of them is satisfied, a contained event $e_j$ is an argument of the containing event $e_i$: (i) The syntactic dependency relation from the predicate of $e_i$ to the predicate of $e_j$ is the clausal complement (*ccomp* or *xcomp*) or clausal subject (*csubj*). (ii) The syntactic dependency relation from the predicate of $e_j$ to the predicate of $e_i$ is copula (*cop*). (iii) All of $e_j$ is contained in an argument of $e_i$ that is labelled with either ARGM-PRP ("purpose") or ARGM-PNC ("purpose not cause"). The syntactic dependency relations are obtained by the Stanford CoreNLP dependency parsing (Manning et al., 2014).

## 4.2 Input Variations

Besides various baseline models, we also explore the influence of different types of input.

**Event-Event Span** We also create the two events' span style input. For the mentioned example, the Event-Event Span style input becomes "'The man who works here {V: tells} me to get the hell out. [SEP] me {V: get} the hell out" – using a special token "[SEP]" for separation.

**Event-Event-SRL** For further checking the influence of the SRL labels, we add the argument

label of the argument where the contained event predicate is in the containing event. We also add a special token "[SRL]" for model recognition. E.g., "The man who works here V: tells me to get the hell out. [SEP] me V: get the hell out [SRL] ARG1."

**Event-Event-SRL-DEP** Based on the Event-Event-SRL Input, we further add the syntactic dependency relation information of the two event predicates. A special token "[DEP]" is added. For instance, "The man who works here V: tells me to get the hell out. [SEP] me V: get the hell out [SRL] ARG1 [DEP] xcomp."

**Marked-Predicate Sentence** Inspired by the success of MarkedBERT (Boualili et al., 2020), we add special marks to emphasize the predicate tokens in the sentence. For example, the sentence "The man who works here tells me to get the hell out." with event predicates "tells" and "get" becomes "The man who works here [V1] tells [\V1] me to [V2] get [\V2] the hell out."

## 4.3 Results and Analysis

Table 3 shows different models' performance based on different inputs. The heuristic rules-based method achieves the highest precision of 97.67%, yet the recall is considerably low at only 34.93%. We observe that adding the event predicate's syntactic dependency and the (semantic) argument label information improves the performance of most of the models. However, using the marked-predicate single sentence as input is more effective than these event-event style inputs. Taking the marked-predicate sentence as input, the BERT and RoBERTa models outperform others.

GPT-2 underperforms compared to discriminative models, indicating its generative architecture struggling with classification tasks such as detecting event dependencies. Similarly, ChatGPT's performance is also unsatisfactory, suggesting that its few-shot learning approach, despite leveraging a powerful pre-trained model, is inadequate for comprehending the dataset. The highest baseline accuracy reaches 82.61%, revealing that EDeR is sufficient for training a good predictor for recognizing the event dependency relations based on the language model's pre-train&fine-tune mechanism.

**Impact of Sentence Length and Predicate Distance.** Figure 4 (left) illustrates the performance of the baseline models across sentences with different lengths, given the input as the marked-predicate sentences. Consistent with human intuition, the

[2]https://chat.openai.com/chat

| Input | Model | P (%) | R (%) | F1(%) | Acc(%) |
|---|---|---|---|---|---|
|  | Majority | 75.50 | 100.00 | 86.04 | 75.50 |
| Sentence+predicates | Rule-based | **97.67 (1)** | 34.93 | 51.46 | 50.08 |
| Event-Event Span | DistilBERT | 82.75 | 90.23 | 86.33 | 78.35 |
|  | BERT | 87.53 | 85.34 | 86.42 | 79.69 |
|  | RoBERTa | 84.71 | 87.53 | 86.10 | 78.58 |
|  | XLNet | 83.61 | 89.60 | 86.5 | 78.82 |
|  | GPT-2 | 85.91 | 86.80 | 86.35 | 79.21 |
|  | ChatGPT | 76.33 | 85.14 | 80.49 | 68.76 |
| Event-Event-SRL | DistilBERT | 84.75 | 87.84 | 86.27 | 78.82 |
|  | BERT | 85.96 | 87.21 | 86.58 | 79.53 |
|  | RoBERTa | 82.89 | **91.16 (3)** | 86.83 | 80.06 |
|  | XLNet | 82.00 | **93.76 (1)** | **87.49 (3)** | 79.69 |
|  | GPT-2 | 86.60 | 85.97 | 86.28 | 79.29 |
|  | ChatGPT | 80.19 | 89.60 | 84.63 | 75.37 |
| Event-Event-SRL-DEP | DistilBERT | 85.48 | 87.53 | 86.49 | 79.29 |
|  | BERT | 85.26 | 89.60 | 87.38 | 80.39 |
|  | RoBERTa | 83.21 | **91.16 (3)** | 87.00 | 80.37 |
|  | XLNet | 83.11 | 91.06 | 86.90 | 79.21 |
|  | GPT-2 | 85.89 | 87.94 | 86.90 | 79.92 |
|  | ChatGPT | 82.01 | 52.60 | 64.09 | 55.39 |
| Marked-predicate Sentence | DistilBERT | 82.13 | **93.14 (2)** | 87.29 | 79.45 |
|  | BERT | **91.30 (2)** | 85.14 | **88.11 (1)** | **82.61 (1)** |
|  | RoBERTa | **89.87 (3)** | 85.85 | **87.81 (2)** | **81.97 (2)** |
|  | XLNet | 88.22 | 86.38 | 87.29 | **80.94 (3)** |
|  | GPT-2 | 85.26 | 89.60 | 87.38 | 80.39 |
|  | ChatGPT | 75.83 | 80.87 | 78.27 | 66.01 |

Table 3: Comparison of the performance on the test set based on varying method and input combinations for the event dependency relation extraction task. The top-3 best results in each column are highlighted. Note: The majority classifier predicts all cases as argument.

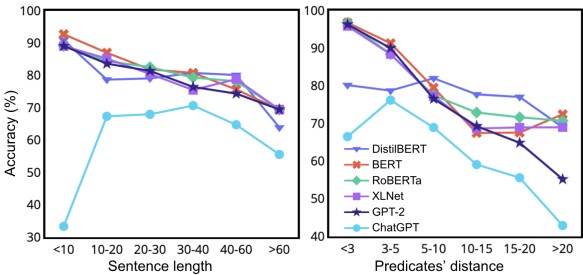

Figure 4: Model performances (accuracy) across different ranges of sentence length (Left) and different ranges of predicates' distance (Right).

performance of all the models (except for Chat-GPT) decreases when the sentence length increases. Longer sentences may contain more complex structures, which are challenging for models to capture. Similarly, Figure 4 (right) shows a decrease in accuracy with an increasing distance (i.e., the number of words) between two predicates, illustrating the models' struggle with distant dependencies. Specifically, few-shot-learning ChatGPT often performs better with moderate sentence length and predicate distance due to challenges with limited information from short sentences and complexity in understanding complex event structures.

# 5 Benefit to Related NLP Tasks

In this section, we show how EDeR data can be used to improve performance on two related NLP tasks.

## 5.1 Event Representation Extraction

As mentioned in Section 3.2.3, in addition to the classification of event dependencies, in the instances where the contained event is not an argument, our data set includes more precise argument spans, which omit parts unique to the contained non-argument event, for containing events. We evaluate the performance of a state-of-the-art Semantic Role Labelling (SRL) system on the resulting refined event extraction task. Our results show that the refined task is harder, and that giving the system access to the dependency classification (annotated or predicted) improves it.

We selected a SOTA SRL system – CRFSRL (Zhang et al., 2022) as the baseline. CRFSRL processes a sentence with a marked event predicate ($p$) by predicting a labelled tree of syntactic dependencies between words, and extracting sub-trees as argument spans. This is illustrated in the top part of Figure 5. Our revised system, CRFSRL with Event Dependency information (**w/ ED**), prunes sub-trees

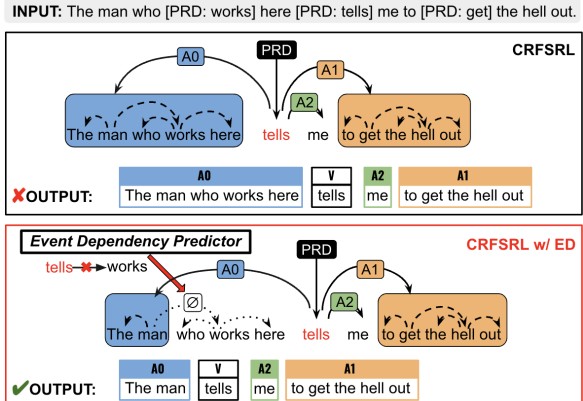

Figure 5: An example of the event representation output by the CRFSRL system (Zhang et al., 2022) (above) and this system equipped with our event dependency relation information (below).

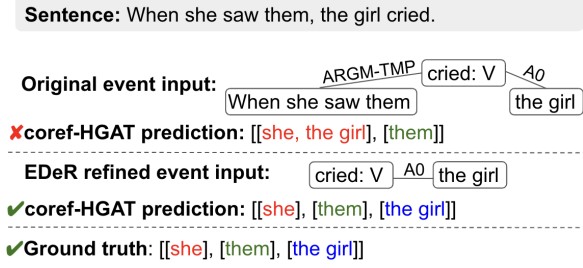

Figure 6: An example shows how EDeR-refined event representations help the coref-HGAT model to get the correct output, unlike when using OntoNotes' original representations.

rooted at non-argument predicates, as shown in the bottom part of Figure 5. In the following, **w/ ED (G)** uses the gold-standard event dependency information from the EDeR annotation, while **w/ ED (P)** uses predicted predicted event dependency relations, from the baseline model with the highest accuracy.

| System | Precision | Recall | F1-score |
|---|---|---|---|
| **Test set:** subset of OntoNotes corresponding to EDeR-test[(a)]. | | | |
| CRFSRL (OntoNotes-trained) | 77.07 | 81.03 | 79.00 |
| **Test set:** EDeR-test. | | | |
| CRFSRL (EDeR-trained) | 74.19 | 78.45 | 76.26 |
| – w/ ED (P) | 74.35 | 78.53 | 76.38 |
| – w/ ED (G) | **76.09** | **78.81** | **77.43** |
| **Test set:** events with refined argument only[(b)]. | | | |
| CRFSRL (EDeR-trained) | 67.34 | 65.73 | 66.53 |
| – w/ ED (P) | 81.14 | 67.97 | 73.97 |
| – w/ ED (G) | **88.11** | **69.51** | **77.71** |

Table 4: Performance of the CRFSRL system (Zhang et al., 2022), and our CRFSRL systems with predicted (**w/ ED (P)**) and annotated (**w/ ED (G)**) event dependency information on the refined event extraction task. (a) For comparison, performance of the CRFSRL system on the original (unrefined) event extraction task. This system is trained on the subset of OntoNotes corresponding to EDeR-train, i.e., with unrefined argument spans, and tested on the subset corresponding to EDeR-test. (b) This is the subset of EDeR-test containing only events with at least one refined argument span (i.e., that differ from the original OntoNotes annotation).

We train both the CRFSRL and CRFSRL w/ ED systems on the training portion of the EDeR data set. Thus, both systems are trained to predict the refined argument spans. For comparison, we also trained a version of CRFSRL on the correspond-

ing subset of the unmodified OntoNotes data set. The evaluation was done using the scripts provided for the CoNLL-2012 shared task[3]. Table 4 summarises the result. CRFSRLs performance on the refined extraction task, compared to the original task, suggests that the system finds it more challenging to model the refined event representations in the EDeR dataset. However, results also show that giving the system access to the event dependency information allows some of that drop to be recovered. The performance gap is much more noticeable on the subset of events in EDeR-test with at least one refined argument span (shown in the bottom part of Table 4). In these cases, using either the baseline model-predicted or human-annotated event dependency feature, CRFSRL w/ ED (P) and (G) both improve significantly (increases of 13.8% and 20.77% in precision, and 7.44% and 11.18% in F1-score) over the CRFSRL system without event dependency information.

These findings underline the importance of event dependency relation information in enhancing the SRL system for event representation extraction, especially for refined events.

## 5.2 Co-reference Resolution

To further investigate whether the refined event representations from EDeR are more reasonable than OntoNotes' original version, we apply them to a downstream task: co-reference resolution (CR).

Given a text, CR aims to identify all mentions that refer to the same entity. We use coref-HGAT (Jiang and Cohn, 2021), a SOTA CR model, initially trained on OntoNotes CR annotations. This model incorporates both event semantic role and word syntactic dependency information in heterogeneous graphs, creating contextualized embed-

---

[3]https://www.cs.upc.edu/ srlconll/

dings to identify co-reference links. However, the model can misinterpret unrelated entities as co-referents. Figure 6 show an example, where "the girl" is wrongly linked to "she". In this example, there is a sentence, "She went to visit them", prior to the input sentence, and "she" in the input sentence actually refers to "She" in this prior sentence. In this case, EDeR's refined event representation removes the distracting potential referent, leading the model to accurately identify "she" and "the girl" as separate entities. While it is also possible that the reduced event representation prunes information that is important for identifying co-referents, our results show that the cases that benefit from it are at least more frequent.

| Input | Precision | Recall | F1-score |
|---|---|---|---|
| OntoNotes Event Reps. (G) | 67.73 | 65.41 | 66.50 |
| EDeR Event Reps. (G) | **69.14** | **65.69** | **67.40** |
| OntoNotes Event Reps. (P) | 64.82 | 63.79 | 64.28 |
| EDeR Event Reps. (P) | **67.79** | **64.42** | **65.96** |

Table 5: Comparison of coref-HGAT model's co-reference resolution performance using annotated (G) and predicted (P) event representations from OntoNotes and EDeR.

We conduct this CR task using the annotations from the same 275 OntoNotes documents and use the CoNLL-2012 official scripts for evaluation.

As Table 5 shows, The EDeR annotated updated event representations (i.e., **EDeR Event Reps. (G)**) boost the coref-HGAT model's precision, recall, and F1-score by 1.41%, 0.28%, and 0.9%, respectively, when compared with the original event representations (i.e., **OntoNotes Event Reps. (G)**). This proves the refined event representations enhance co-reference resolution, demonstrating their validity.

Furthermore, the coref-HGAT model with CRF-SRL w/ ED (P) predicted updated events (i.e., **EDeR Event Reps. (P)**) also reveals enhanced precision, recall, and F1-score by 2.97%, 0.63%, and 1.68%, compared with the model with CRF-SRL predicted OntoNotes original events (i.e., **OntoNotes Event Reps. (P)**), as the last two lines in Table 5) shows. Despite model prediction errors, these results are even comparable to the performance of using OntoNotes Event Reps. (G). This highlights the feasibility of applying the CR model more widely, by substituting the original annotated event inputs with our predicted refined event representations.

## 6 Conclusion

In this paper, we introduced EDeR, a high-quality human-annotated event dependency relation dataset. We presented a thorough description of the dataset construction process followed by a detailed analysis. We implement various language models and one unsupervised rule-based method, to establish benchmark performance against the dataset. The experimental results of these competitive baselines, along with the evident improvements in some related NLP tasks such as semantic role labelling and cor-reference resolution, demonstrate the utility and benefit of the EDeR dataset. We hope that EDeR facilitates future research in uncovering potential relations among events, thereby enriching the understanding of our linguistic world.

## Limitations

As described, for our EDeR dataset, the *argument* label can be further categorized into two fine-grained classes: *required argument* and *optional argument*. We apply the introduced baseline models and inputs to this three-way classification task. The experimental results and some case study details are in A.2. Overall, compared with the binary classification on argument/non-argument, the performance is significantly dropped - the highest accuracy is 70.79% and the ChatGPT few-shot learning results are much worse. It demonstrates that the three-way classification is more challenging for our current baseline models. Such a significant performance gap will direct our further investigation.

## Ethical Considerations

We annotate the proposed EDeR dataset based on OntoNotes 5.0, without copyright constraints for academic uses. During the human annotation procedure (cf. Section 3), the annotators, quality inspectors, and the linguistic expert are only required to label factual information (i.e., the dependency relations between events and refined event representations). The annotators and quality inspectors are all anonymous and the annotation does not involve any personally sensitive information.

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

## A  Appendix

### A.1  Statistics of Documents for Data Collection

OntoNotes contains semantic role-formatted event representations, as the OntoNotes example in Figure 2 shows. We randomly sampled 275 documents from seven genres: broadcast news (bn), magazine (mz), newswire (nw), pivot corpus (pt), telephone conversation (tc), broadcast conversation (bc), and web data (wb). Data statistics of the 275 raw documents are shown in Table 6. The number of sampled documents and the separation of them into training, development and test sets under each genre follows their initial distributions in the OntoNotes dataset.

### A.2  Fine-grained Event Dependency Relation Extraction

As described, for our EDeR dataset, the argument label can be further categorized into two more fine-grained classes: *required argument* and *optional argument*. Likewise, the non-argument label can be further separated into the two classes *condition* and *independent*. We apply the baseline models and inputs also to these 3-way and 4-way classification tasks.

The 3-way classification results of the baseline language models taking various types of inputs are shown in Table 7. First, the Event-Event-SRL-DEP input achieves overall better performance than others in this task. Besides the useful information provided by the syntactic dependencies as discussed in Section 4.4, another possible reason is that in the EDeR dataset, the predicates of over 95.79% of the

Figure 7: Case study for fine-grained event dependency relation extraction.

events labelled as required arguments are located within the numbered arguments (ARG0–ARG5) of their containing events. Second, like in the binary classification task, the BERT and RoBERTa models outperform the others. RoBERTa achieves the best overall precision, recall, F1-score and accuracy results, which reveals its better relation reasoning capacity. Third, the overall precision and recall results for the "required argument" label tend to be higher in comparison to other labels, likely as a result of its preponderance in the dataset.

Finally, the overall performance on both the 3-way and 4-way classification tasks is lower than for binary classification – the highest accuracy achieved are 70.79% and 69.69%, respectively – indicating that these tasks are harder.

### A.2.1  Case Study

We identify two phenomena from the Event-Event-SRL-DEP based RoBERTa model for the fine-grained event dependency relation extraction:

**Syntactic Dependency Parsing Errors**  Some incorrect (three-way) classification results may be because of the misleading syntactic dependency parsing results. As shown in the first case of Figure 7, the off-the-shelf dependency parser wrongly assigns syntactic dependency relation between the two event predicates "work" and "have", which could affect the model prediction.

**Event Argument Labelling Errors**  Furthermore, as the second case of Figure 7 shows, the semantic labels provided by OntoNotes for the event predicate "had" is incorrect (e.g., "the group" is missing as the ARG0). It may prevent the model from accurately predicting the event dependency relation types.

|  | bn | mz | nw | pt | tc | bc | wb | overall |
|---|---|---|---|---|---|---|---|---|
| # documents | 104 | 7 | 107 | 31 | 7 | 3 | 16 | 275 |
| # documents-train | 90 | 5 | 89 | 24 | 5 | 1 | 13 | 227 |
| # documents-dev | 7 | 1 | 9 | 4 | 1 | 1 | 2 | 25 |
| # documents-test | 7 | 1 | 9 | 3 | 1 | 1 | 1 | 23 |
| avg # sentences per doc | 6.2 | 48.9 | 14.4 | 40.7 | 71.7 | 203.8 | 59.3 | 24.9 |
| avg # words per doc | 139.7 | 1599.7 | 418.7 | 657.0 | 1089.4 | 4045.8 | 1509.3 | 543.4 |
| avg # events per doc | 23.6 | 202.1 | 58.3 | 148.7 | 362.1 | 1058.6 | 268.5 | 93.5 |

Table 6: Statistics of the documents (under each genre and all) that we sampled from the OntoNotes dataset for annotation: number of documents and the number of documents split into different sets (train, development and test), and average number of sentences, words and events per document.

| Input | Model | Precision (%) | | | | Recall (%) | | | | F1(%) | Accuracy(%) |
|---|---|---|---|---|---|---|---|---|---|---|---|
| | | required | optional | non-arg. | overall | required | optional | non-arg. | overall | | |
| ① | DistilBERT | 76.06 | 52.04 | 55.06 | 61.05 | 81.31 | 45.11 | 56.49 | 60.97 | 60.90 | 64.80 |
| | BERT | 82.90 | 58.48 | 54.62 | 65.33 | 85.69 | 44.02 | 67.21 | 65.64 | 64.92 | 69.13 |
| | RoBERTa | 83.47 | 54.01 | 59.41 | 65.63 | 83.33 | 54.89 | 58.44 | 65.56 | 65.59 | 69.06 |
| | XLNet | 80.46 | 50.71 | 56.39 | 62.52 | 83.16 | 48.37 | 55.84 | 62.46 | 62.47 | 66.46 |
| | GPT-2 | 76.41 | 52.01 | 56.68 | 61.70 | 82.32 | 45.65 | 56.49 | 61.49 | 61.49 | 65.43 |
| | ChatGPT | 47.63 | 31.08 | 32.08 | 36.93 | 52.53 | 47.55 | 37.59 | 35.19 | 32.32 | 39.65 |
| ② | DistilBERT | 75.04 | 51.50 | **64.68** | 63.74 | 84.51 | 55.98 | 42.21 | 60.90 | 61.41 | 65.98 |
| | BERT | 84.59 | 54.55 | 57.89 | 65.68 | 83.16 | 53.80 | 60.71 | 65.89 | 65.77 | 69.21 |
| | RoBERTa | 83.75 | 58.13 | 56.37 | 66.08 | 84.18 | 50.54 | 64.61 | 66.44 | 66.08 | 69.69 |
| | XLNet | 81.66 | 59.27 | 56.20 | 65.71 | 84.68 | 44.29 | 69.16 | 66.04 | 65.28 | 69.21 |
| | GPT-2 | 76.12 | 54.75 | 58.25 | 63.04 | 82.66 | 47.01 | 58.44 | 62.70 | 62.73 | 66.46 |
| | ChatGPT | 54.55 | 32.45 | 26.92 | 37.97 | 65.66 | 46.74 | 2.27 | 38.22 | 34.02 | 44.77 |
| ③ | DistilBERT | 77.27 | 53.93 | 60.23 | 63.81 | 81.82 | 55.98 | 50.65 | 62.82 | 63.15 | 66.77 |
| | BERT | 81.51 | 56.42 | 60.70 | 66.21 | 85.35 | 51.36 | 61.69 | 66.13 | 66.12 | 69.76 |
| | RoBERTa | 83.20 | **59.38** | 59.04 | **67.21** | **85.86** | 52.45 | 63.64 | **67.31** | **67.15** | **70.79** |
| | XLNet | **86.27** | 53.68 | 58.07 | 66.01 | 82.49 | 55.43 | 60.71 | 66.21 | 66.08 | 69.37 |
| | GPT-2 | 78.58 | 53.42 | 56.57 | 62.86 | 82.15 | 46.74 | 60.06 | 62.99 | 62.82 | 66.54 |
| | ChatGPT | 55.07 | 26.18 | 27.39 | 36.21 | 19.19 | 37.77 | 47.25 | 34.74 | 31.36 | 31.39 |
| ④ | DistilBERT | 73.07 | 51.91 | 61.57 | 62.18 | 81.31 | 48.10 | 53.57 | 60.99 | 61.40 | 64.96 |
| | BERT | 85.26 | 53.80 | 58.66 | 65.91 | 81.82 | 50.00 | 68.18 | 66.67 | 66.13 | 69.29 |
| | RoBERTa | 84.32 | 56.93 | 56.23 | 65.83 | 84.18 | 51.36 | 62.99 | 66.17 | 65.89 | 69.53 |
| | XLNet | 84.75 | 55.34 | 54.41 | 64.83 | 80.47 | 46.47 | 70.13 | 65.69 | 64.78 | 68.11 |
| | GPT-2 | 77.00 | 50.00 | 55.11 | 60.70 | 45.38 | 57.79 | **79.46** | 60.88 | 60.74 | 64.33 |
| | ChatGPT | 37.68 | 28.42 | 25.00 | 32.39 | 17.51 | **66.03** | 11.33 | 33.22 | 27.54 | 31.86 |

Table 7: Comparison of the performance for the 3-way event dependency relation classification task. Input data types are: ① Event-Event Span; ② Event-Event-SRL; ③ Event-Event-SRL-DEP and ④ Marked-predicate Sentence. The precision and recall results for particular labels are presented as columns under "required", "optional" and "non-argument". The best results in each column are highlighted. The overall precision, recall and F1-score results are macro-averaged.

## A.3 Implementation Settings

We employ the `distilbert-base-cased`, `bert-large-cased`, `roberta-large`, `xlnet-large-cased` and `gpt2-large` in HuggingFace's Transformer Library (Wolf et al., 2020) as representing the DistilBERT, BERT, RoBERTa, XLNet and GPT-2 baseline models in the experiment. For the fairness of comparison, all the models use the same optimization method as AdamW (Loshchilov and Hutter, 2017) which is adopted with an initial learning rate of $1e-5$ and a batch size of 4 and a number of epochs of 4 during the finetuning process. The model with the highest F1-score on the development set is selected.

## A.4 Annotation Instructions

An event is composed of a verb and arguments, where the verb indicates the event's action and the arguments represent the subject, object and so on of the verb. This task requires you to determine whether, in a pair of events (Event 1, Event 2) from the same sentence, Event 2 is an argument or a condition of Event 1, and if it is an argument, whether it is required or optional.

• **Required argument**: Event 2 is required for the verb of Event 1 to be complete and meaningful.
For example,
**Sentence**: Jenny tries to stop them from ruining her family.
**Event 1**: Jenny V: tries to stop them from ruining her family.
**Event 2**: Jenny V: stop them from ruining her family. ("V: ..." marks the verb of each event.)
**Explanation**: Event 2 is a required argument. If it is removed, Event 1 becomes just "Jenny tries (to)", which is incomplete and not meaningful (to "try to", one has to try to do something).
Another example of the required argument:
**Sentence**: Krishna suggests Ramesh to send his liquor to town.
**Event 1**: Krishna V: suggests Ramesh to send his liquor to town.
**Event 2**: Ramesh V: send his liquor to town.
**Explanation**: Event 2 is a required argument. If it is removed, Event 1 becomes "Krishna suggests Ramesh", which is not meaningful (to suggest to someone, one must should suggest something).
The third example of the required argument:
**Sentence**: "The stock price is unstable." Mr. Jones says.
**Event 1**: "The stock price is unstable." Mr. Jones

V: says.
**Event 2**: The stock price V: is unstable.
**Explanation**: Event 2 is a required argument. If it is removed, Event 1 would become "Mr. Jones says", which is incomplete and not meaningful (one must say something, "something" is missing if Event 2 is omitted).

• **Optional argument**: Event 2 is an argument of the verb of Event 1, but Event 2 only provides additional or clarifying information about Event 1 (for example, Event 2 may indicate the purpose or goal of Event 1). Event 1 is still syntactically complete and meaningful, although it may not keep the exact same meaning, if Event 2 is removed.
For example,
**Sentence**: A power failure with the docking door forces Brodski to go EVA to fix it.
**Event 1**: Brodski V: go EVA to fix it.
**Event 2**: Brodski V: fix it.
**Explanation**: Event 2 is an optional argument, indicating the purpose of Event 1. Removing it leaves Event 1 "Brodski go(es) EVA", which is still a syntactically complete and meaningful clause.
Another example of the optional argument relation:
**Sentence**: It is a good opportunity, as eventually turned out to be.
**Event 1**: It V: is a good opportunity, as eventually turned out to be.
**Event 2**: eventually V: turned out to be.
**Explanation**: Event 2 is an optional argument, adding information about the timing of Event 1. Removing it leaves Event 1 "It is a good opportunity", which is still a syntactically complete and meaningful clause.

• **Condition**: Event 2 describes a condition for the occurrence of Event 1.
For example,
**Sentence**: Joe will hate her if she tells the truth.
**Event 1**: Joe will V: hate her if she tells the truth.
**Event 2**: she V: tells the truth.
**Explanation**: Event 2 is a condition of Event 1; it is not an argument.
Another example,
**Sentence**: Should problems arise, we will seek professional help.
**Event 1**: Should problems arise, we will V: seek professional help.
**Event 2**: Should problems V: arise.
**Explanation**: Event 2 is a condition of Event 1; it is not an argument.

• **Neither an argument nor a condition**: Event

2 is not an optional or required argument, or condition of Event 1. It is an independent/separate event, which happens to be mentioned in the same sentence. Like an optional argument or condition, removing Event 2 still leaves Event 1 complete and meaningful. The difference to Event 2 being an optional argument is that when Event 2 is not an argument it does not add any information/clarification about the verb of Event 1; it may still add some information about one of the arguments of Event 1.

For example,

**Sentence**: Julia, who has fallen in love with Barnabas, discovers his dalliance with Maggie.

**Event 1**: Julia, who has fallen in love with Barnabas, V: discovers his dalliance with Maggie.

**Event 2**: Julia, who has V: fallen in love with Barnabas

**Explanation**: Event 2 is unrelated to Event 1; it is a separate event that happens to share the same subject. Whether Julia falls in love with Barnabas or not does not affect whether, how, or when, she discovers his dalliance with Maggie.

Another example:

**Sentence**: It increases the discount rate it offers to known customers.

**Event 1**: It V: increases the discount rate it offers to known customers.

**Event 2**: the discount rate it V: offers to known customers.

**Explanation**: Event 2 is unrelated to Event 1; it is a separate event that only describes the object (the discount rate) of Event 1. Event 2 does not affect the verb of Event 1 "increases".

The third example of neither an argument nor a condition:

**Sentence**: After an opening-credits montage of the major players demonstrating their abilities, the story begins.

**Event 1**: After an opening-credits montage of the major players demonstrating their abilities, the story V: begins.

**Event 2**: an opening-credits montage of the major players V: demonstrating their abilities.

**Explanation**: Event 2 is a separate/independent event that happens before Event 1. Event 2 only adds some description of the opening-credits montage; it does not add information about Event 1 ("The story begins"). The ordering between the two events does not make Event 2 an argument of Event 1.