# OpenReview forum: "EDeR: Towards Understanding Dependency Relations Between Events"
_EMNLP/2023/Conference — EMNLP 2023 Main_

### Official Review · Reviewer_R3vw · 2023-07-25

**Soundness:** 4

**Excitement:**

4: Strong: This paper deepens the understanding of some phenomenon or lowers the barriers to an existing research direction.

**Missing References:**

I'm aware of several datasets that are distinct in their aims from the present work but that are worth citing as relevant general background:
- The Basic dataset from the IARPA BETTER program (see [here](https://ir.nist.gov/better/)) contain event structures with eventive arguments (so-called "referred events"). NB: To my knowledge, there is no publication associated with the Basic datasets specifically, but see [this paper](https://aclanthology.org/2022.lrec-1.384/) for a description of the program and the related Abstract datasets.
- "Richer Event Description: Integrating event coreference with temporal, causal and bridging annotation" (O'Gorman et al., 2016) has received a good amount of attention and contains several different types of event-event relations.
- "Decomposing and Recomposing Event Structure" (Gantt et al., 2022) annotates the English Web Treebank for event parthood relations as part of its event structure dataset.

**Paper Topic And Main Contributions:**

This paper presents the Event Dependency Relation (EDeR) dataset, which annotates binary argument relations on close to 12,000 pairs of predicates from a subset of documents from the OntoNotes dataset. Relations between predicates are assigned one of four labels: for eventive arguments, one predicate may be semantically **required** by the other or else may be **optional**; and for eventive *non-arguments*, one predicate may nonetheless express a **condition** on the other predicate or else may be fully **independent** of it. The paper also presents baseline models for relation classification on EDeR in both fine-tuned (using standard pretrained LMs) and few-shot settings (using ChatGPT), with experiments showing performance variation across different input representations. Results from additional experiments argue that incorporating event dependency information into training for related tasks (semantic role labeling and coreference resolution) may boost performance on those tasks.

**Questions For The Authors:**

A. How was the total number of documents for annotation (275) determined? Was this just based on resource constraints or was there some other principle behind it?

B. It is counterintuitive to me that the modified event structures would boost performance on entity coreference resolution. If I understand correctly, by pruning dependency parses based on the event relations, isn't it possible that you are dropping spans that participate in a gold coreference chain? Or does the model still have access to these spans, albeit not as part of the dependency parse? I feel like I must be misunderstanding something here.

C. The authors note that "For the analysis of the dataset and experiments in the remainder of this paper, we treat the two non-argument labels as one" (L253-5). Why was this done? Among non-arguments, the distinction between a condition and a wholly independent event is semantically very salient.

**Reasons To Accept:**

- The dataset seems to be fairly carefully constructed and likely to be of interest to a significant subset of the NLP community, including folks working on event semantics and on information extraction. The fact that it builds on an existing widely used benchmark (OntoNotes) is also a plus.
- The experiments in general seem to accomplish most of what one expects of experiments in a resource paper — namely, establishing an accessible set of baselines and demonstrating the utility of the resource for existing tasks.

**Reasons To Reject:**

I do not in general have significant methodological concerns with the paper; most of my concerns relate to presentation and certain points of clarity (see **Questions for The Authors** and **Typos, Grammar, Style, and Presentation Improvements**), though I'll raise one point about annotator agreement here. While the authors' description of the data collection procedure inspires confidence with respect to the dataset's quality (notably, the fact that the 5% sampled annotations had 95% accuracy with respect to the authors' judgments), no standard agreement metrics are presented on the full subset of the data that is redundantly annotated. This does make it somewhat difficult to get a full sense for its quality.

**Reproducibility:**

4: Could mostly reproduce the results, but there may be some variation because of sample variance or minor variations in their interpretation of the protocol or method.

**Reviewer Confidence:**

4: Quite sure. I tried to check the important points carefully. It's unlikely, though conceivable, that I missed something that should affect my ratings.

**Typos Grammar Style And Presentation Improvements:**

Typos / Grammar:

In general, the paper would benefit from a careful proofread. Below are a sampling of suggested edits:
- 142-3: "Several event-event relations have been proposed for decades" -> (e.g.) "Various event relation ontologies have been proposed over the years."
- 179: "an optional argument of, or a condition of" -> "an optional argument of, a condition of"
- 332: hence -> and so
- 394-6: this description of the rule-based classification method is rather awkward. Please rephrase.
- 529: feature -> features
- figure 6 caption: help -> helps

Presentation:
- Majority class baseline accuracy really should be explicitly presented (even if only in text) for comparison with the results in Table 3 (though I believe it's inferrable from Table 2).
- Table 3 should also include F1 scores
- For clarity, it would be good to have explicit examples of the "Event-Event-SRL" and "Event-Event-SRL-DEP" input variations presented in 4.2. (This is already done for the other two variants.)
- An analysis of model performance broken down by label, possibly including some qualitative evaluations, would (for my money) be more interesting and informative than the present analysis based on predicate distance and sentence length (though this analysis is still worthwhile).

---

> ### Author Rebuttal · Authors · 2023-08-29
>
> We appreciate the reviewer's insightful feedback. In response to the concerns highlighted in "Reasons To Reject", "Questions For The Authors", and the suggestions for "Presentation Improvement", we present the following explanations and adjustments:
>
> ### Reasons To Reject:
>
> I do not in general have significant methodological concerns with the paper; most of my concerns relate to presentation and certain points of clarity (see Questions for The Authors and Typos, Grammar, Style, and Presentation Improvements), though I'll raise one point about annotator agreement here. While the authors' description of the data collection procedure inspires confidence with respect to the dataset's quality (notably, the fact that the 5% sampled annotations had 95% accuracy with respect to the authors' judgments), no standard agreement metrics are presented on the full subset of the data that is redundantly annotated. This does make it somewhat difficult to get a full sense for its quality.
>
> **Author response:** Regarding the evaluation of the quality of EDeR, we have some statistics that are not presented in the paper due to the page limit:
>
> 1. Among the 4771 samples (40.2% of the total 11852 cases) from EDeR that have received the three Quality Inspectors' annotations, only 31 (0.65%) of them received three different labels from the three QIs.
>
> 2. The authors randomly sampled 417 cases that received at least four annotations; the accuracy of the annotated labels is 90.65% (378/417).
>
> 3. The authors also randomly sampled 205 cases from the cases that received a single annotation (from one annotator), the accuracy is 83.41% (171/205).
>
> We will include these statistics in the revision.
>
> ### Questions For The Authors:
>
> * **Question:** How was the total number of documents for annotation (275) determined? Was this just based on resource constraints or was there some other principle behind it?
>
>   **Answer:** The number of documents we sampled corresponds to the number of event pairs we aim to annotate. Based on existing research in event relation extraction, we believe a minimum of 10k samples is essential for fine-tuning a pre-trained language model. For reference, the BERT baseline model already attains an accuracy of 82.61%. This sampling helps us gauge the potential benefits of these event dependency relations. We're considering annotating more documents in the future to further enhance the dataset.
>
> * **Question:** It is counterintuitive to me that the modified event structures would boost performance on entity coreference resolution. If I understand correctly, by pruning dependency parses based on the event relations, isn't it possible that you are dropping spans that participate in a gold coreference chain? Or does the model still have access to these spans, albeit not as part of the dependency parse? I feel like I must be misunderstanding something here.
>
>   **Answer:** It is possible that our refined event representations drop spans that participate in a gold coreference chain, but such instances are fairly rare. As illustrated in Figure 6, "she" in the example sentence does not refer to "the girl". There is a sentence prior to the example sentence in the document: "She went to visit them." "she" in the example sentence actually refers to "She" in this prior sentence. We shorten the span of the event with the verb "cried" by removing "when she saw them", which breaks the dependency path between "she" and "the girl" then successfully corrects the incorrect co-referring chain between them.
>
> * **Question:** The authors note that "For the analysis of the dataset and experiments in the remainder of this paper, we treat the two non-argument labels as one" (L253-5). Why was this done? Among non-arguments, the distinction between a condition and a wholly independent event is semantically very salient.
>
>   **Answer:** Thank you for pointing it out. The mentioned sentence is indeed misplaced and will be removed in the revision. We've presented data set statistics for the two non-argument labels separately, in table 2. As for experiments, we've also executed both three-way (required argument, optional argument, and non-argument) and four-way (required argument, optional argument, condition, and neither an argument nor a condition) classifications utilizing the baseline models. The results of the three-way classification and case studies can be found in Appendix A.2. In our revision, we'll incorporate the four-way classification results. The classification accuracy has seen a decline of about 15% in comparison to the binary classification models.
>
> ### Missing References, Typos, and Presentation Improvement:
>
> **Overall response:** We are grateful for your literature recommendations. Especially in the third paper, where the proposed "parthood event-event relation" aligns closely with our event dependency relations and offers significant inspiration. We also thank you for identifying typos and suggesting improvements in the representation. We will cite the missing references, provide more technical details, and conduct more rounds of proofreading to ensure that grammar issues are fixed.
>
> Specifically, addressing the representation concerns you've highlighted:
>
> * Majority class baseline accuracy really should be explicitly presented (even if only in text) for comparison with the results in Table 3 (though I believe it's inferrable from Table 2).
>
>   **Author response:** We'll incorporate the baseline accuracy details you mentioned into Table 3 for a more explicit comparison in the revision.
>
> * Table 3 should also include F1 scores.
>
>   **Author response:** We will include the F1-score into Table 3 in the next edition.
>
> * For clarity, it would be good to have explicit examples of the "Event-Event-SRL" and "Event-Event-SRL-DEP" input variations presented in 4.2. (This is already done for the other two variants.)
>
>   **Author response:** Due to page constraints, we omitted examples for both "Event-Event-SRL" and "Event-Event-SRL-DEP". Here's an illustrative example for each:
>
>   * **Event-Event-SRL:** "The man who works here {V: tells} me to get the hell out. [SEP] me {V: get} the hell out [SRL] ARG1."
>   * **Event-Event-SRL-DEP:** "The man who works here {V: tells} me to get the hell out. [SEP] me {V: get} the hell out [SRL] ARG1 [DEP] xcomp."
>
>    We aim to include such examples in the revision for clarity.
>
> * An analysis of model performance broken down by label, possibly including some qualitative evaluations, would (for my money) be more interesting and informative than the present analysis based on predicate distance and sentence length (though this analysis is still worthwhile).
>
>   **Author response:** We will include the P/R/F1 results for particular labels, respectively, for the 3-way and 4-way classifications in the revision. Similar to our approach with the 3-way classification results (can be found in Appendix A.2.1), we will also do some qualitative evaluations, like case studies, for binary classification.

---

### Official Review · Reviewer_9FEu · 2023-08-04

**Soundness:** 4

**Excitement:**

3: Ambivalent: It has merits (e.g., it reports state-of-the-art results, the idea is nice), but there are key weaknesses (e.g., it describes incremental work), and it can significantly benefit from another round of revision. However, I won't object to accepting it if my co-reviewers champion it.

**Paper Topic And Main Contributions:**

The paper introduces human-annotated dataset EDeR. The dataset extracts event dependency information based on a sample set of documents and provides refined semantic role labeled event representations.

**Reasons To Accept:**

The motivation is very clear. Paper presents a thorough description of annotation process followed by well-designed experiments and analysis.

**Reasons To Reject:**

While the paper introduces new dataset along with the solid explanation of data construction procedure, the novelty of the paper is not enough.

**Reproducibility:**

5: Could easily reproduce the results.

**Reviewer Confidence:**

4: Quite sure. I tried to check the important points carefully. It's unlikely, though conceivable, that I missed something that should affect my ratings.

---

> ### Author Rebuttal · Authors · 2023-08-29
>
> We are grateful for the reviewer's perceptive feedback. Although the reviewer hasn't posed any specific questions to us, we would like to address the points raised under the "Reasons To Reject" section.
>
> ### Reasons To Reject:
>
> While the paper introduces new dataset along with the solid explanation of data construction procedure, the novelty of the paper is not enough.
>
> **Author response:** We'd like to highlight several aspects that underscore the novelty of our paper:
>
> 1. **Shift in focus from entities to events:** For decades, much of the previous literature in relation extraction domain has been concentrated on relationships between entities rather than events. Our paper shifts significantly from this norm by bringing to the forefront the importance of events and their intricate interrelationships.
>
> 2. **Broadening the understanding of event relations:** While existing research on event relations has primarily explored causal, temporal, and hierarchical relations, our study embarks on a previously underexplored domain: the syntactic and semantic dependencies between events. Such an exploration addresses the gaps left by current event-relation systems, thereby broadening our overall comprehension of event dynamics.
>
> 3. **Highlighting overlooked dependencies:** Our work is pioneering in that it emphasizes the often-overlooked interdependence of events. The prevailing research has mostly treated events as syntactically and semantically independent, potentially leading to incomplete or even erroneous interpretations of narratives. By exploring these deeper dependencies, our study proposes a more holistic approach to event relation extraction.
>
> 4. **Real-world application and impact:** By understanding the dependencies between events, we can not only interpret textual nuances more accurately but also reduce potential misinterpretations. This paper, in particular, shows how recognizing event dependency can innovatively alter the interpretation of the narrative. Also, the experimental results demonstrate the improvement of such relations in some popular NLP tasks such as semantic role labeling and co-reference resolution. Its impact on the development of the NLP community is significant.

---

### Official Review · Reviewer_HYy2 · 2023-08-11

**Soundness:** 4

**Excitement:**

4: Strong: This paper deepens the understanding of some phenomenon or lowers the barriers to an existing research direction.

**Paper Topic And Main Contributions:**

This paper introduces a human-annotated Event Dependency Relation dataset (EDeR) which explores the interdependency between events. Annotators annotate argument relationships between the event pairs. The quality of the dataset is assured by a multi-level qualification-based annotation procedure. The experiment results demonstrate the utility and benefit of the EDeR dataset.

**Questions For The Authors:**

1. Table 3. Why the recall score of ChatGPT is only 52.6% under Event-Event-SRL-DEP setting?
2. Table 4. What is the meaning of “All update events”? Is it “Refined events only” plus “unchanged events”? If so, why does “Refined events only” achieve better results in F1 scores, since the training set is smaller than “All update events”?
3. Line 554-556. Is it more likely that “she” and “the girl” refer to the same entity?

**Reasons To Accept:**

1. The focus on addressing independence between two events is quite interesting and makes a lot of sense.
2. This paper explores the conditional and dependency relationships between events, which is helpful in some related NLP tasks.
3. The dataset has high-quality annotation results under a reasonable annotation procedure.
4. The experiments are well set and show positive results.


**Reasons To Reject:**

1. Experimental results do not show the effectiveness of fine-grained classes.
2. Pherpas lacks Inter-annotator Agreement to better demonstrate the quality of EDeR.

**Reproducibility:**

3: Could reproduce the results with some difficulty. The settings of parameters are underspecified or subjectively determined; the training/evaluation data are not widely available.

**Reviewer Confidence:**

3: Pretty sure, but there's a chance I missed something. Although I have a good feel for this area in general, I did not carefully check the paper's details, e.g., the math, experimental design, or novelty.

---

> ### Author Rebuttal · Authors · 2023-08-29
>
> We appreciate the reviewer's thoughtful feedback. In response to the "Reasons To Reject" and the "Questions For The Authors" presented by the reviewer, we would like to address them as follows:
>
> ### Reasons To Reject:
>
> Experimental results do not show the effectiveness of fine-grained classes.
>
> Pherpas lacks Inter-annotator Agreement to better demonstrate the quality of EDeR.
>
> **Author response:** Current fine-tuned language models exhibit a 10% lower accuracy for fine-grained classes compared to binary ones. Given that binary classes already sufficiently cater to downstream tasks such as co-reference resolution, our immediate focus is on enhancing the predictor's performance. Nevertheless, evaluating the effectiveness of fine-grained classes is firmly on our future work agenda.
>
> Regarding the demonstration of the quality of EDeR, we have some statistics that are not presented in the paper due to the page limit:
>
> 1. Among the 4,771 samples (40.2% of the total 11852 cases) from EDeR that have received the three Quality Inspectors' annotations, only 31 (0.65%) of them received three different labels from the three QIs.
>
> 2. The authors randomly sampled 417 cases that received at least four annotations; the accuracy of the annotated labels is 90.65% (378/417).
>
> 3. The authors also randomly sampled 205 cases from the cases that received a single annotation (from one annotator), the accuracy is 83.41% (171/205).
>
> We will include these statistics in the revision.
>
> ### Questions For The Authors:
>
> * **Question:** Table 3. Why the recall score of ChatGPT is only 52.6% under Event-Event-SRL-DEP setting?
>
>   **Answer:** The potential reasons:
>
>    1. **Distribution of DEP features:** A significant portion of the test set data, specifically over 70% labeled as "argument", has the DEP feature classified as "none". More than half of the "non-argument" data also has the feature as "none". Given our application of few-shot learning on ChatGPT, this overlap might confuse the model, complicating its inference abilities.
>
>    2. **Utility of specific dependency relations:** While dependency relations like "ccomp" and "xcomp" assist in detecting required and optional argument relations, they primarily enhance precision. For instance, event pairs with "ccomp" or "xcomp" relations are likely to exhibit event dependency relations, but the reverse isn't necessarily true. This observation is evident from our proposed rule-based approach. For unsupervised methods, or models like few-shot learning-based ChatGPT, while precision is satisfied, the recall might be significantly lower.
>
>    3. **Comparison of SRL and DEP features:** The SRL feature operates differently from the DEP feature. Primarily, the semantic role of argument events tends to be numbered arguments. For instance, over 78% of cases labeled as "argument" in the test set possess the SRL feature categorized as either "ARG1" or "ARG2". In contrast, this ratio drops to 39% for "non-argument" cases. This distinction might explain why the SRL feature, when used solely, yields superior performance.
>
>
> * **Question:** Table 4. What is the meaning of “All update events”? Is it “Refined events only” plus “unchanged events”? If so, why does “Refined events only” achieve better results in F1 scores, since the training set is smaller than “All update events”?
>
>   **Answer:** Yes, "all update events" consist of both "refined events only" and "unchanged events". We utilized the EDeR training set events to train the CRFSRL model, aiming to predict "all updated events" and "refined events only" separately. The results in Table 4 reflect comparisons made by the CRFSRL model using the same training set, but differing test sets. Specifically, "all updated events" refers to the complete EDeR test set events, while "refined events only" pertains to just the refined events from the EDeR test set. We apologize for any confusion and will ensure clarity in the subsequent edition.
>
>
> * **Question:** Line 554-556. Is it more likely that “she” and “the girl” refer to the same entity?
>
>   **Answer:** "she" in the example sentence does not refer to "the girl". There is a sentence prior to the example sentence in the document: "She went to visit them." "she" in the example sentence actually refers to "She" in this prior sentence.

---

### Meta-Review · Area_Chair_bgvU · 2023-09-18

**Recommendation:** 5

**Metareview:**

The paper introduces a human-annotated Event Dependency Relation dataset.

The three reviewers agree on the strong soundness of the work (4) and moderate to strong excitement (4-3-4). The reviewers have provided several suggestions for improvement and questions that should be incorporated in the final version of the manuscript if accepted. Among these, I find the following particularly relevant:
- Inter-annotator Agreement to better demonstrate the quality of EDeR - and any other quality statistics (such as as the one described in the rebuttal).
- Typos, grammar, style, and presentation improvements

Although I agree that a statement such as " the novelty of the paper is not enough" is unsupported, I believe that including in the paper the novelty highlights summarized in the rebuttal could improve the final version.

---

### Decision · Program_Chairs · 2023-10-07

**Decision:**

Accept-Main

**Comment:**

The paper introduces a human-annotated Event Dependency Relation dataset.

The three reviewers agree on the strong soundness of the work (4) and moderate to strong excitement (4-3-4). The reviewers have provided several suggestions for improvement and questions that should be incorporated in the final version of the manuscript if accepted. Among these, I find the following particularly relevant:
- Inter-annotator Agreement to better demonstrate the quality of EDeR - and any other quality statistics (such as as the one described in the rebuttal).
- Typos, grammar, style, and presentation improvements

Although I agree that a statement such as " the novelty of the paper is not enough" is unsupported, I believe that including in the paper the novelty highlights summarized in the rebuttal could improve the final version.